# Ultrathin Flexible Encapsulation Materials Based on Al_2_O_3_/Alucone Nanolaminates for Improved Electrical Stability of Silicon Nanomembrane-Based MOS Capacitors

**DOI:** 10.3390/mi15010041

**Published:** 2023-12-24

**Authors:** Zhuofan Wang, Hongliang Lu, Yuming Zhang, Chen Liu, Haonan Zhang, Yanhao Yu

**Affiliations:** 1Key Laboratory for Wide Band Gap Semiconductor Materials and Devices of Education Ministry, School of Microelectronics, Xidian University, Xi’an 710071, China; zhuofanxidian@sina.com (Z.W.); zhangym@xidian.edu.cn (Y.Z.);; 2Department of Materials Science and Engineering, Southern University of Science and Technology, Shenzhen 518055, China; yuyh@sustech.edu.cn

**Keywords:** ultrathin flexible encapsulation, Al_2_O_3_/alucone, silicon nanomembrane, metal-oxide-semiconductor capacitors, electrical stability

## Abstract

Ultrathin flexible encapsulation (UFE) using multilayered films has prospects for practical applications, such as implantable and wearable electronics. However, existing investigations of the effect of mechanical bending strains on electrical properties after the encapsulation procedure provide insufficient information for improving the electrical stability of ultrathin silicon nanomembrane (Si NM)-based metal oxide semiconductor capacitors (MOSCAPs). Here, we used atomic layer deposition and molecular layer deposition to generate 3.5 dyads of alternating 11 nm Al_2_O_3_ and 3.5 nm aluminum alkoxide (alucone) nanolaminates on flexible Si NM-based MOSCAPs. Moreover, we bent the MOSCAPs inwardly to radii of 85 and 110.5 mm and outwardly to radii of 77.5 and 38.5 mm. Subsequently, we tested the unbent and bent MOSCAPs to determine the effect of strain on various electrical parameters, namely the maximum capacitance, minimum capacitance, gate leakage current density, hysteresis voltage, effective oxide charge, oxide trapped charge, interface trap density, and frequency dispersion. The comparison of encapsulated and unencapsulated MOSCAPs on these critical parameters at bending strains indicated that Al_2_O_3_/alucone nanolaminates stabilized the electrical and interfacial characteristics of the Si NM-based MOSCAPs. These results highlight that ultrathin Al_2_O_3_/alucone nanolaminates are promising encapsulation materials for prolonging the operational lifetimes of flexible Si NM-based metal oxide semiconductor field-effect transistors.

## 1. Introduction

Wearable and implantable electronics require reliable encapsulation layers to prevent ions and gas in the surrounding environment from deteriorating their electronic properties [1,2,3]. Recently, the lifetimes of flexible transistors based on ultrathin channels, such as silicon nanomembrane (Si NM) and polysilicon, have been prolonged by applying ultrathin flexible encapsulation (UFE) layers of less than one micrometer thick [4,5,6]. Compared with traditional encapsulation materials, such as titanium and bulk ceramics, emerging UFE materials have more advantages, namely scalable thickness and mechanical properties that enable seamless contact between devices and curved, dynamic surfaces [7,8]. For hermetic enclosure materials with low water vapor transmission rates (WVTRs), UFE layers are typically generated through the atomic layer deposition (ALD) of aluminum oxide (Al_2_O_3_) films or multilayered inorganic structures [1,9,10]. As applied, the compressive or tensile strain increases to the crack onset strain (COS) and undesirable cracks start to form and then propagate in ALD-Al_2_O_3_ films, which significantly undermine the protective function of the film [11]. Molecular layer deposition (MLD) of organic films, such as aluminum alkoxide (alucone) of tens of nanometers thick, enhances the mechanical stability of the UFE layer because the organic films (e.g., alucone) can be introduced as interlayers of the inorganic film (e.g., Al_2_O_3_) [3,12,13]. Notably, the COS of alucone films (~14%) is far higher than that of ultrathin Al_2_O_3_ (~2%) [14,15]. Using several dyads (one Al_2_O_3_ layer and one alucone layer) to generate Al_2_O_3_/alucone nanolaminates on plastic substrates has been proven to sustain extremely low WVTRs and maintain mechanical stability under bending strains [12].

A daunting challenge is to realize encapsulated transistors exhibiting strain-insensitive barrier performance under mechanical bending strains, which means that electrical properties, such as threshold voltage and mobility, must remain stable or exhibit negligible variations [16,17,18,19,20]. However, unencapsulated devices have ultrathin channels with fracture limits (~1% for Si NM), and thus the critical parameters usually change with the external strain [21,22]. On the one hand, to achieve flexible, high-performance, high-speed, low-power dissipation metal oxide semiconductor transistors, high-quality gate oxide is required to realize strong gate control capabilities. Low sub-threshold swing (SS), high current switching ratio (I_on_/I_off_), high transconductance (g_m_), and controllable threshold voltage are critically needed. To obtain higher yields and robust device reliability, investigations should focus on high-quality gate oxide and packaging layers, which can increase yield and improve device reliability. Variating these parameters with strain is essential for enhancing electrical stability and reliability in wearable, foldable, stretchable, and implantable devices. On the other hand, the investigations on encapsulated thin film transistors lack mechanism-related evidence to verify the effectiveness of improvement on electrical characteristics at bending strains. A key and practical strategy is to use Si NM-based metal oxide semiconductor capacitors (MOSCAPs) with UFE layers to investigate the effect of bending strains on electrical performance. The dynamic changes of UFE layers on flexible substrates with different bending radii have been thoroughly investigated, although the stability of interfacial characteristics of encapsulated devices under mechanical strains has yet to be confirmed. 

In this work, we generated 3.5 dyads of Al_2_O_3_/alucone (11 nm/3.5 nm) on Si NM-based MOSCAPs using ALD/MLD. The electrical properties of encapsulated and unencapsulated MOSCAPs under bending strains were comprehensively analyzed to demonstrate the improved electrical stability of encapsulated MOSCAPs. The device fabrication and encapsulation methods are described in Section 2. Capacitance versus voltage (C–V) and current density versus voltage (J–V) curves of the MOSCAPs in the flat state were obtained (Section 3) to investigate the effects of bending strains on electrical properties (Section 4). Critical parameters that reflect the stability of the dielectric layer and interface at dielectric/Si NM, namely the maximum capacitance (C_max_), minimum capacitance (C_min_), gate leakage current density (J_g_), and hysteresis voltage (V_hy_) of capacitors with and without encapsulation were further analyzed. In addition, the effective oxide charge (N_eff_), oxide trapped charge (N_ot_), interface trap density (D_it_), and frequency dispersion (Freq. (ω)) were analyzed to determine the effectiveness of the deposited ultrathin encapsulation structure. The results indicated that Al_2_O_3_/alucone nanolaminates improved the mechanical strain insensitivity of MOSCAPs without affecting the original electrical characteristics.

## 2. Device Fabrication and Encapsulation Methods

Figure 1 illustrates the layout of the MOSCAP encapsulated with 3.5 dyads of Al_2_O_3_/alucone. Device fabrication began with a silicon-on-insulator (SOI) wafer with a 200 nm device layer and a 150 nm buried oxide (BOX) layer. The wafer was diced into 2 × 2 cm squares, and then arrays of hollow cubes were patterned (depth: 226 nm) on the surface of the sample using photolithography and reactive ion etching (RIE). The sample was ultrasonically cleaned in acetone and soaked in 49% hydrofluoric acid (HF) solution until the BOX layer was fully dissolved. Si NM on the handling wafer was immediately transferred into the chamber, where Ti/Au (10/170 nm) metal stacks were deposited on suspended Si NM via e-beam evaporation (EBE) to serve as the cathode electrode. The multilayer structure was transferred from the handling wafer onto the adhesive SU-8-coated polyethylene terephthalate (PET) substrate (3 × 3 cm, thickness: 250 μm) via a flipping transfer scheme, followed by ultraviolet (UV) light exposure to crosslink SU-8. Conventional fabrication steps were performed on the Si NM/cathode, including mesa isolation, dielectric deposition, anode deposition, and dielectric etching. The dielectric layer was ultrathin Al_2_O_3_/HfO_2_ (5 nm/5 nm), and the anode was Ti/Au (10/170 nm). The flexible MOSCAPs on PET substrate were instantly taped on an uploading chuck and transmitted into the ALD/MLD chamber, where 3.5 dyads of Al_2_O_3_/alucone nanolaminates were deposited at 120 °C. The conditions of Al_2_O_3_ deposition were 100 cycles of trimethylaluminum (TMA) pulse for 0.1 s, N_2_ purging for 10 s, H_2_O pulse for 0.1 s, N_2_ purging for 20 s, 10 cycles for alucone deposition, TMA pulse for 0.2 s, N_2_ purging for 10 s, ethylene glycol (EG) pulse for 0.5 s, and N_2_ purging for 20 s. The thickness of the film encapsulating the MOSCAP was confirmed using a spectroscopic ellipsometer. Then, encapsulation of the backside of the PET substrate was performed to complete the procedure. Bending tests of as fabricated and Al_2_O_3_/alucone-encapsulated MOSCAPs used concave and convex molds, as shown in Figure 2. C–V and I–V measurements were carried out by Keysight B1500A semiconductor parameter analyzer (Santa Rosa, CA, USA), which connected to the Cascade Microtech EPS 150 manual probe station (Beaverton, OR, USA). All the measurements were performed at room temperature and in a dark box. The bending radii and related strains were benchmarked against the previous research on Si NM based flexible devices.

## 3. C–V and J–V Characteristics of MOSCAPs in the Pristine State

C–V and J–V curves of the MOSCAPs in their original states are shown in Figure 3a,b. It is notable from the optical images (Figure 3a) that the anode area of MOSCAPs is yellow before the encapsulation process. It then significantly changes into purple after depositing 3.5 dyads Al_2_O_3_/alucone films, and the deposition area can be identified through the arrays of etching holes on the functional layer of Si NM, in which the color of the holes has changed. These changes indicated that the structures of MOSCAPs were seamlessly protected after conformal encapsulation. The two types of MOSCAPs had the same C_max_ of 0.275 μF/cm^2^, and the similar slopes in the depletion zone indicated that the interfacial characteristics were unchanged after encapsulation. In addition, the flatband voltage (V_fb_) shifted from −0.76 V to −0.70 V after encapsulation, indicating that the electrical properties remained stable after Al_2_O_3_/alucone encapsulation using ALD/MLD. Moreover, the gate leakage current densities of encapsulated and unencapsulated MOSCAPs at 0.5 V were 3.12 × 10^−9^ A/cm^2^ and 3.11 × 10^−9^ A/cm^2^, respectively (Figure 3b, red dashed lines). Hence, the electrical characteristics of unencapsulated and Al_2_O_3_/alucone-encapsulated MOSCAPs in the pristine state were used as baselines for further comparisons of strain-related characteristics. A total of 20 devices were prepared from the same fabrication methods of samples, the test results were repeatable, and the trends on curves were consistent. Although there are slight differences in values, the trends are consistent. Therefore, we selected significant and representative results for discussion.

## 4. Electrical Characteristics at Different Bending Radii

### 4.1. Comparison of Measured C–V Curves at Different Bending Strains

Figure 4 (left) shows the C–V curves of the gate voltage, which were obtained by sweeping from depletion to inversion with Al_2_O_3_/alucone-encapsulated Si NM-based MOSCAPs bent and fixed on molds through polyimide (PI) adhesive tapes. The same flexural tests were performed on unencapsulated MOSCAPs, and the results are shown in Figure 4 (right). The surfaces of the molds were grinded and polished to minimize the contact resistance between the plastic substrate and the bending apparatus. The top surface strain (ε), mainly defined by the thickness of the substrate, was calculated as follows [23].
(1)$$\varepsilon =(t_{sub}+t_{device})/2R$$
where t_sub_ and t_device_ are the thickness of the PET substrate and MOSCAPs (including Si NM, metal stacks, and dielectric), respectively, and R denotes the flexural radii of the molds bent inwardly to radii of 85 mm and 110.5 mm and outwardly to radii of 77.5 mm and 38.5 mm. The calculated ε was −0.15%, −0.11%, +0.16%, and +0.33% (where − represents compression strain from inward bending, + represents tensile strain from outward bending). The measured C_max_ decreased, and the shapes of the C–V curves deteriorated to different degrees as the inward and outward bending strains varied. N_eff_ and N_ot_ related mechanisms under bending strains reduced C_max_ by different magnitudes. In addition, the presence of D_it_ under mechanical strains varied the slope from depletion to inversion, indicating that “stretch-out” on the C–V curve changed with the applied voltage under bending strains. 

### 4.2. Comparative Analysis of C_max_, C_min_, J_g_, V_hy_

At an unbent state, 3.5 dyads Al_2_O_3_/alucone encapsulation does not deteriorate the electrical performance. It is meaningful to investigate the variations on the dielectric layer at bending strains. The maximum capacitance denotes oxide capacitance, and the minimum capacitance is influenced by semiconductor capacitance. Figure 5a shows the ε-dependent changes in C_max_ normalized by the C_max_ of the MOSCAPs in the pristine state (ΔC_max_/C_max,o_). At the compressive strains of −0.15% and −0.11%, the C_max_ of the unencapsulated MOSCAP decreased by 0.6% and 0.9%, respectively, whereas the C_max_ of the Al_2_O_3_/alucone-encapsulated MOSCAP increased by 0.1% and 0.9%, respectively. The C_max_ of the encapsulated device was relatively stable at the tensile strains of +0.16% and +0.33%, whereas the C_max_ of the unencapsulated device markedly decreased at the high tensile strain of +0.33%. This trend was attributed to the encapsulation multilayers on the ultrathin silicon channel, which minimized the propagation of local strain into the MOSCAP. Hence, under mechanical strain, the Al_2_O_3_/alucone-encapsulated MOSCAP was more stable than the unencapsulated MOSCAP. Specifically, the C_max_ of the former deteriorated by only 7%, while that of the latter deteriorated by 31%. Figure 5b depicts the effect of ε on the C_min_ of the two types of MOSCAPs. The C_min_ of the Al_2_O_3_/alucone-encapsulated MOSCAP was extremely stable and increased by less than 3% at ε = −0.15%, −0.11%, and +0.16%. At the high strain of +0.33%, the C_min_ of the encapsulated MOSCAP decreased by 0.6% with a C_max_ decline of 7.1%. In contrast, the C_min_ of the unencapsulated MOSCAP in the investigated four bending states significantly decreased. The results of C_max_ and C_min_ indicated that the Si NM MOSCAP with the Al_2_O_3_/alucone nanolaminate was robust and that encapsulation substantially enhanced the stability of the MOSCAP during operation.

Figure 6a shows the current density versus voltage curves of encapsulated and unencapsulated MOSCAPs under different strains (ε). The changes in J_g_ divided by J_g_ in the unbent state (ΔJ_g_/J_g,o_) are comparatively analyzed in Figure 6b. For the Al_2_O_3_/alucone encapsulated MOSCAP, J_g_ decreased by 12.4% and then 14.8% as the compressive strain increased but minimally increased by 2.3% and then 3.7% as the tensile strain increased. The decrease in J_g_ under compressive strains was attributed to the Al_2_O_3_/alucone encapsulation structure providing more current to the current pathway in the top-gate configuration, namely the inwardly bent direction. However, the J_g_ of the unencapsulated MOSCAP increased whether under compressive or tensile strains. These results highlighted that Al_2_O_3_/alucone encapsulation stabilized the J_g_ of MOSCAPs and the certain layouts of encapsulation structures were insensitive to strains.

Figure 7a shows the normalized capacitance versus voltage curves of unencapsulated and encapsulated MOSCAPs in the unbent and four bent states, which were obtained by sweeping in the forward and reverse directions. Figure 7b shows the changes in V_hy_ at different bending strains divided by V_hy_ in the unbent state (ΔV_hy_/V_hy,o_). The results showed that the V_hy_ of the encapsulated MOSCAP significantly increased as the strain increased. We inferred that high-energy ion bombardment during the ALD/MLD process generates mobile ions in the Al_2_O_3_/alucone encapsulated MOSCAP, leading to a larger and counterclockwise fluctuating V_hy_ under bending strains compared with that of the unencapsulated MOSCAP. However, the V_hy_ values of the two types of MOSCAPs were small, both of which were 0.02 V, indicating that the gate electrode of the encapsulated MOSCAP in the pristine shielded this effect. The results of N_ot_ mentioned below verified this inference. Nevertheless, this instability can be eliminated through annealing, as demonstrated for other ultrathin channel-based thin-film transistors (TFTs) [24]. On the other hand, the investigation focuses on the relative change. The possible pre-existing defects in the oxide layer are included as the baseline for encapsulated and unencapsulated MOSCAPs at the original state. Thus, the V_hy_-related changes are more likely to be defined as process-related defects, gradually influencing the V_hy_ at compressive and tensile strains.

### 4.3. Comparision of N_eff_, N_ot_, D_it_, Freq.(ω) under Bending Strains

Figure 8a,b shows the ε-dependent changes in the effective oxide charge (N_eff_) and oxide trapped charge (N_ot_) divided by the corresponding values in the unbent state. N_eff_ and N_ot_ were calculated as follows [25].
(2)$$\left\{\begin{array}{l} N_{eff}=\frac{(V_{fb\_theor.}-V_{fb})\times C_{\max}}{q\times A}\\ N_{ot}=\frac{V_{hy}\times C_{\max}}{q\times A} \end{array}\right.$$
where V_fb_theor._ is the theoretical flatband voltage, q is the electron charge, and A is the active area of MOSCAPs. The results showed that the N_eff_ of the encapsulated MOSCAP was stable and changed by up to 12% under bending strains, leading to a slight negative shift in V_fb_. At the high tensile stain of +0.33%, the N_eff_ of the unencapsulated MOSCAP was smaller than that of the encapsulated MOSCAP by 42% (Figure 8a, yellow bar), which was attributed to the decreased C_max_ of the unencapsulated MOSCAP. These results indicated that without encapsulation, the interface between the oxide and ultrathin Si channel was unstable. The N_ot_ values of the encapsulated and unencapsulated MOSCAPs in the unbent state were 0.33 × 10^11^ cm^−2^ and 0.32 × 10^11^ cm^−2^, respectively. However, these values (representing the abundant oxide trapped charges near the interface of the oxide and ultrathin Si channel) changed as the bending strain changed, which increased the V_hy_ of the encapsulated MOSCAP under strain. This trend highlighted the necessity to monitor the chamber while depositing multilayers on flexible oxide-based TFTs. 

D_it_ is a critical parameter to evaluate the robustness of the interfacial characteristics. In this work, D_it_ values were extracted by analyzing the ideal and measured capacitance, which varied with the voltage, of encapsulated and unencapsulated MOSCAPs in each bent state using the Terman method. The values of D_it_ versus the energy level were numerically derived as follows [26]
(3)$$\left\{\begin{array}{l} D_{it}=D_{it}(V_{s})=\frac{C_{ox}}{q^{2}A}\times \frac{d(V_{g}-V_{g,id})}{dV_{s}}=\frac{C_{ox}}{q^{2}A}\times \frac{d\Delta V_{g}}{dV_{s}}\\ E_{c}-E=kT\times \ln\left(\frac{N}{n_{i}}\right)-qV_{s} \end{array}\right.$$
where V_s_ is the surface potential, C_ox_ is the oxide capacitance, V_g_ is the applied voltage, and V_g, id_ is the gate voltage with the same ideal capacitance as the measured capacitance with V_g_. N is channel doping concentration. n_i_ is intrinsic carrier concentration. Moreover, E is the energy level of interface trap states, while E_c_ denotes the conduction band. 

Figure 9a shows the D_it_ versus Ec−E curves of encapsulated and unencapsulated MOSCAPs under different strains (ε). At the unbent state, the distribution of D_it_ values monotonically decreased with the energy level for encapsulated and unencapsulated MOSCAPs. With changes in bending strains, the slope of the D_it_ distribution for unencapsulated MOSCAPs is gradually smoother than the unbent state. However, the D_it_ distribution for encapsulated MOSCAPs is stable at the applied bending strains. The increase in D_it_ values significantly demonstrates that the interfacial characteristics deteriorated under bending strains. Yet, 3.5 dyads of Al_2_O_3_/alucone ultrathin encapsulation films stabilized the interfacial characteristics at bending strains. Specifically, to compare the relative changes in D_it_, the energy level of E_c_−E = 0.35 eV was selected as the reference because the changes at that energy level tended to be steady on the D_it_ distribution. Extracted D_it_ values divided by the D_it_ values in the unbent state (D_it/_D_it,o_) at various bending strains are shown in Figure 9b. The D_it_ values for MOSCAPs without encapsulation are more susceptible to being affected by the tensile strains. The results showed that the D_it_ of the unencapsulated MOSCAP at E_c_−E = 0.35 eV is 0.96 × 10^12^ eV^−1^ cm^−2^ of unbent state and increased 93.1% to achieve 1.86 × 10^12^ eV^−1^ cm^−2^ at +0.33%. Moreover, the variations are as significant as +9.7% to +13.8% for the other three bending strains. In contrast, the D_it_ of the encapsulated MOSCAP at E_c_−E = 0.35 eV maintains steadily within the variations of −6.4% to +11.4%. The strain-related D_it_ distributions for Al_2_O_3_/alucone-encapsulated and unencapsulated MOSCAPs are comparatively analyzed to prove that the ultrathin encapsulation multilayers are promising to stabilize the interfacial characteristics at bending strains. On the one hand, it may be attributed to the Al_2_O_3_/alucone nanolaminates having the ability to prevent ions in the air going into the dielectric/Si NM interface from deteriorating the electrical performances. On the other hand, the ultrathin Al_2_O_3_/alucone balanced the residual strain caused by etching holes on Si NM, which improved the interfacial characteristics compared to unencapsulated MOSCAPs.

Figure 10a shows the monotonic decrease in the C_max_ with a frequency range of 1 to 100 kHz for unencapsulated and Al_2_O_3_/alucone-encapsulated MOSCAPs in the unbent and bent states. The extracted values of the frequency dispersion (Freq.(ω)_o_) of the unencapsulated and encapsulated MOSCAPS in the pristine state (Figure 10a, dotted line) were 7.8%/Dec and 8.9%/Dec, respectively. However, Freq.(ω) normalized by Freq.(ω)_o_ changed with the applied bending strain, as shown in Figure 10b. The Freq.(ω)/Freq.(ω)_o_ values of the unencapsulated MOSCAP showed that Freq.(ω) increased twofold at ε = +0.33% but decreased by more than half at ε = −0.15%. In contrast, the Freq.(ω)/Freq.(ω)_o_ values of the encapsulated MOSCAP were stable owing to the protective function of the Al_2_O_3_/alucone multilayers. In addition, under compressive and tensile strains, the Freq.(ω)/Freq.(ω)_o_ values of the encapsulated MOSCAPs were significantly smaller than those of the unencapsulated MOSCAP, which indicated that the ultrathin encapsulation structure promoted electrical stability. Moreover, the minimal change in Freq.(ω) with bending strain demonstrated that Al_2_O_3_/alucone encapsulation stabilized N_eff_. It verified that the 3.5 dyads Al_2_O_3_/alucone nanolaminates efficiently promoted the stability and reliability of the dielectric layer for Si NM-based flexible MOSCAPs. 

Based on the observations of changes in electrical properties for 3.5 dyads Al_2_O_3_/alucone ultrathin film-encapsulated and unencapsulated MOSCAPs, the results evoke robust evidence on the mechanisms between Si NM-based MOSCAPs and ultrathin inorganic–organic encapsulating materials. The encapsulated Si NM-based MOSCAPs could withstand more compressive and tensile strains than unencapsulated MOSCAPs. From the device aspect, the ultrathin multilayered structures provide a tortuous path compared to a single layer, preventing the external molecules or gases from penetrating the dielectric layer. This ensures gate oxide reliability when the flexible MOSCAPs endure external bending strains. Compared to the unencapsulated MOSCAPs, the stability of gate leakage current, effective oxide charges, interfacial characteristics, and frequency dispersion are sufficient to evidence that proposal. From the fabrication aspect, the ALD/MLD procedure truly has more advantages in precise thickness control. However, the ion bombardment with high energy causes process-related issues in this work, and the variations in hysteresis voltage change with dynamic bending strains. This requires more consideration when designing anode stacks with efficient thickness and the scalable layout of encapsulated devices. From the fracture strain aspect, the mechanical bending strain leads to microcracks in the multilayered device structures. The physical microcracks influence the electrical property with complex mechanisms. Here, the electro-mechanical analysis on 3.5 dyads Al_2_O_3_/alucone nanolaminate-encapsulated MOSCAPs highlighted the strain-related electrical parameters such as oxide charges and interface trapped charges, potential tools providing theoretical evidence and practical results to electronic design automation (EDA) designers. It is helpful to further devise strain-insensitive, flexible transistors with reliable encapsulation structures. 

## 5. Conclusions

In summary, we successfully deposited ultrathin Al_2_O_3_/alucone nanolaminates on Si NM-based MOSCAPs. In addition, we systematically analyzed the effect of mechanical strain on the electrical properties of unencapsulated and encapsulated MOSCAPs. The two types of MOSCAPs in the pristine had similar electrical properties, namely C_max_, C_min_, V_hy_, J_g_, N_eff_, N_ot_, D_it_, and Freq.(ω). However, as the compressive and tensile strains increased, the electrical properties of the Al_2_O_3_/alucone-encapsulated MOSCAP were more stable than those of the unencapsulated MOSCAP. The strain-related mechanisms on electrical stability for 3.5 dyads Al_2_O_3_/alucone nanolaminate-encapsulated MOSCAPs were comprehensively analyzed and comparatively discussed in this work. These findings highlight that the Al_2_O_3_/alucone ultrathin encapsulation structure is a robust design for achieving high strain insensitivity of Si NM-based metal oxide semiconductor field-effect transistors and integrated circuits with long operational lifetimes in wet and curved application scenarios. The further development of ultrathin inorganic–organic encapsulation for high-performance, wearable, and implantable electronics needs efforts to excavate scientific issues such as bringing the stretchable designs as a form factor with ultrathin inorganic–organic encapsulation.

## Figures and Tables

**Figure 1 micromachines-15-00041-f001:**
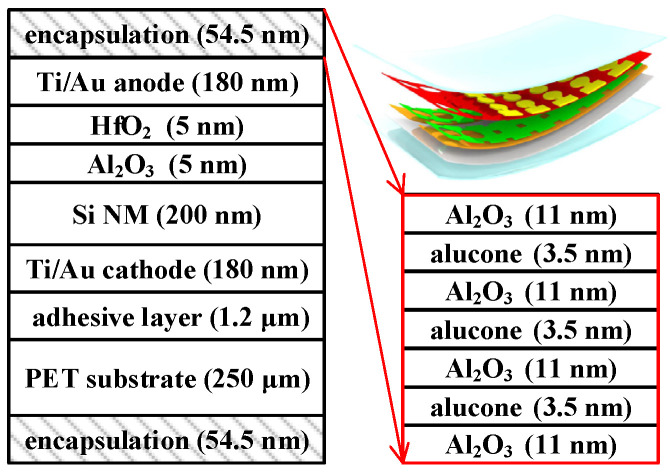
Schematic illustration of the device structure. (The thickness is not drawn to scale).

**Figure 2 micromachines-15-00041-f002:**
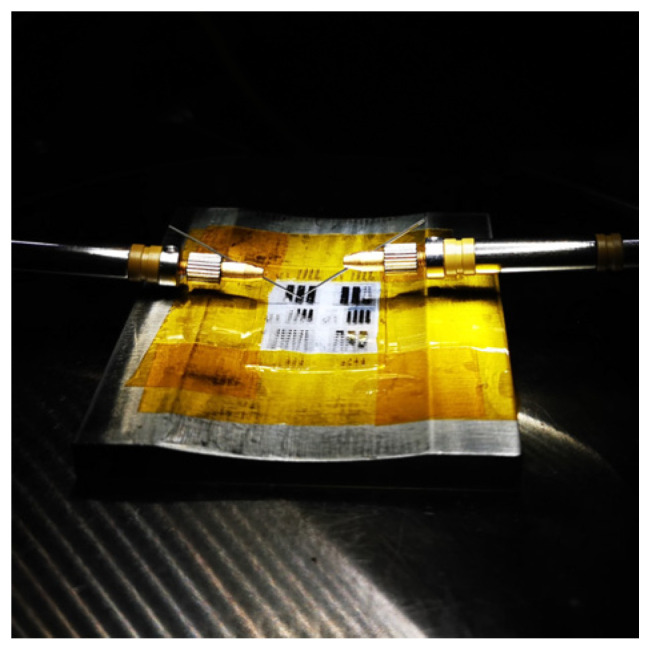
Optical image of electrical measurements for MOSCAPs bent on concave mold.

**Figure 3 micromachines-15-00041-f003:**
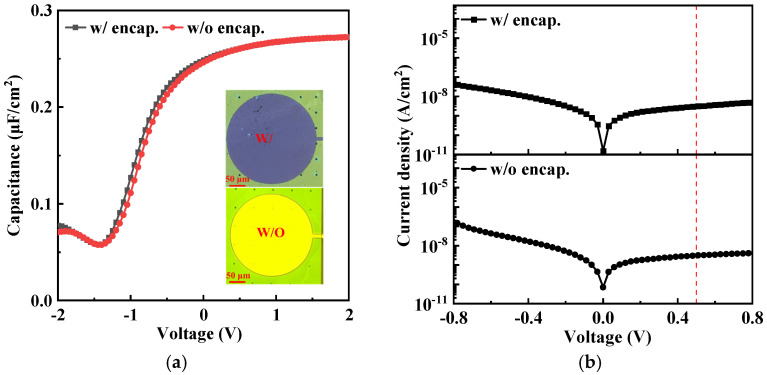
(**a**) C–V and (**b**) J–V curves of MOSCAPs with and without Al_2_O_3_/alucone encapsulation in the flat state. Insets: microscopic images of the top gate for MOSCAPs with and without encapsulation. Scale bar: 50 μm.

**Figure 4 micromachines-15-00041-f004:**
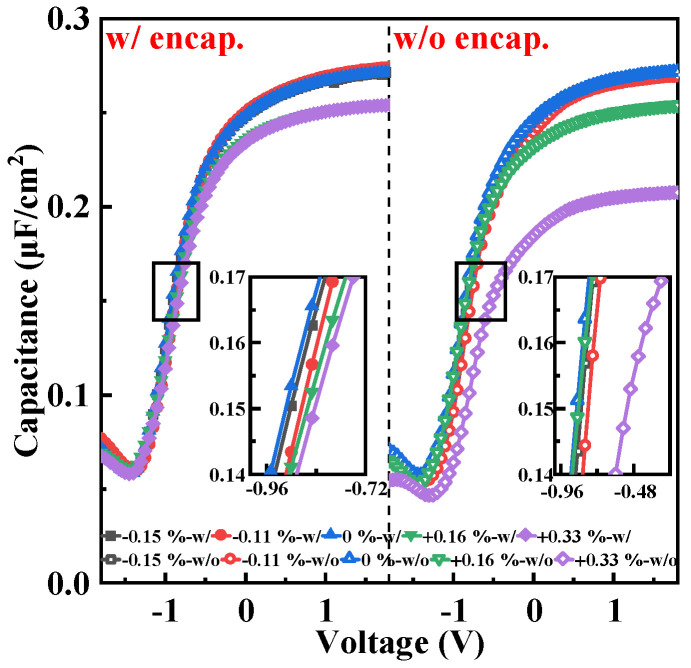
Experimental capacitance versus voltage curves of Al_2_O_3_/alucone-encapsulated (**left**) and unencapsulated (**right**) MOSCAPs in the unbent state and under inward/outward bending strains. The insets show an enlarged view of the voltage sweep from −1 to −0.72 V (**left**) and from −1 to −0.24 V (**right**).

**Figure 5 micromachines-15-00041-f005:**
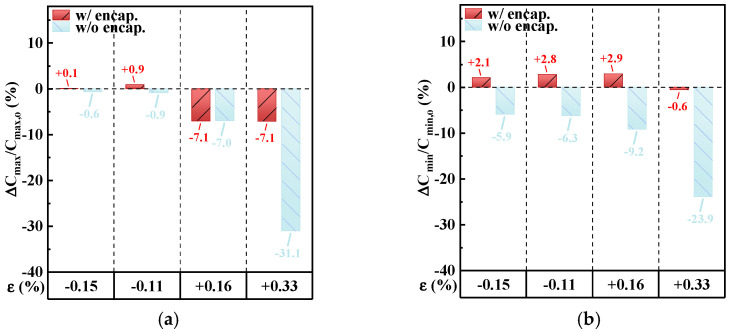
Changes in the (**a**) maximum capacitance (ΔC_max_/C_max,o_) and (**b**) minimum capacitance (ΔC_min_/C_min,o_) at different inward and outward bending strains.

**Figure 6 micromachines-15-00041-f006:**
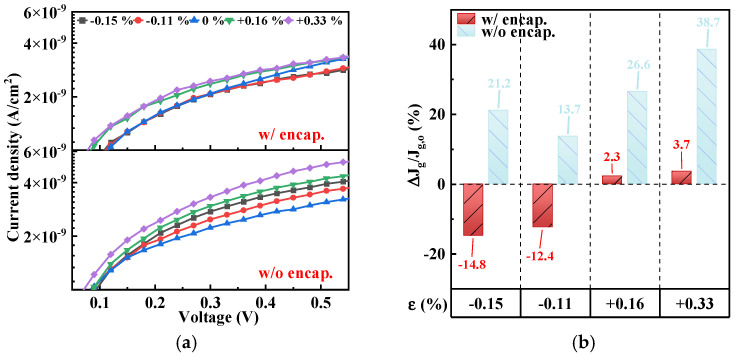
(**a**) Current density versus voltage curves at different bending strains. (**b**) Changes in the gate leakage current density (ΔJ_g_/J_g,o_) at different inward and outward bending strains.

**Figure 7 micromachines-15-00041-f007:**
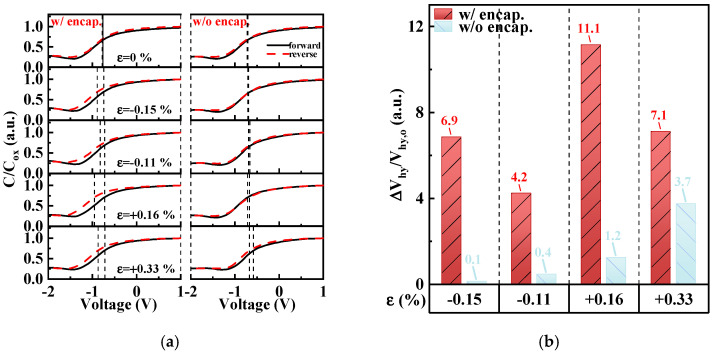
(**a**) Normalized capacitance (C/C_ox_) versus voltage curves swept from forward and reverse directions at various bending strains. The dashed line indicates the position of V_fb_, which corresponds to the flatband capacitance (C_fb_). (**b**) Changes in the hysteresis voltage (ΔV_hy_/V_hy,o_) at various bending strains.

**Figure 8 micromachines-15-00041-f008:**
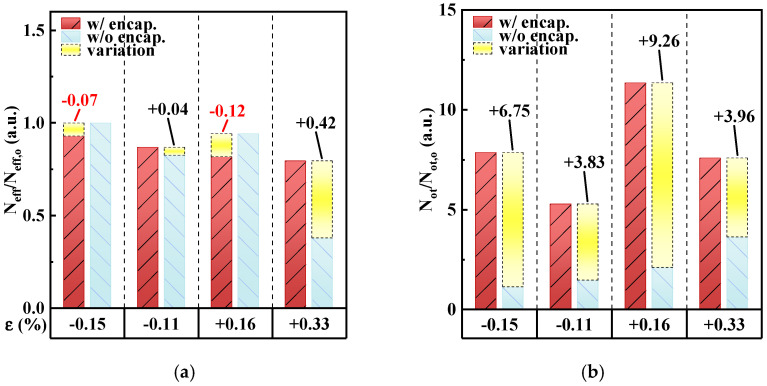
(**a**) Effective oxide charge (N_eff_/N_eff,o_) and (**b**) oxide trapped charge (N_ot_/N_ot,o_) at various bending strains.

**Figure 9 micromachines-15-00041-f009:**
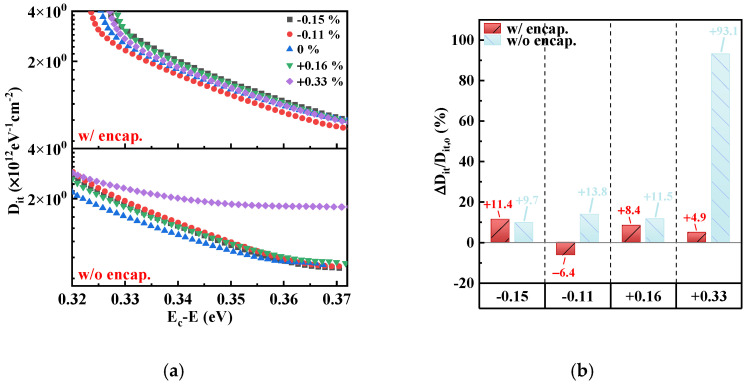
(**a**) Interface trap density (D_it_) versus energy level of E_c_−E at different bending strains. (**b**) D_it_ corresponds to E_c_−E at 0.35 eV divided by D_it_ in the unbent state (D_it_/D_it,o_) at different bending strains.

**Figure 10 micromachines-15-00041-f010:**
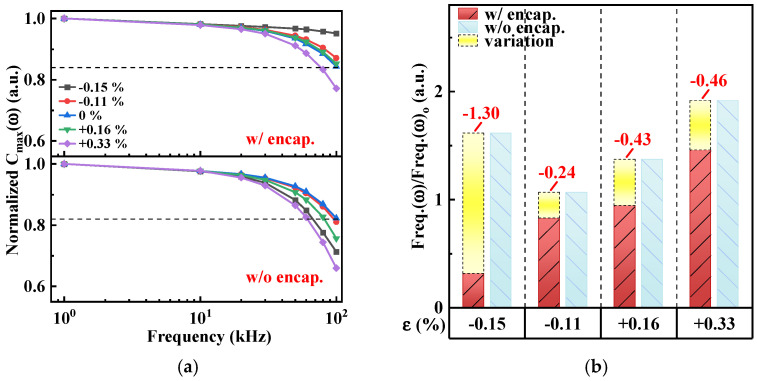
(**a**). Normalized C_max_ versus frequency curves at different bending strains. (**b**) Frequency dispersion normalized by Freq.(ω)_o_ (Freq.(ω)/Freq.(ω)_o_) at different bending strains.

## Data Availability

Data are contained within the article.

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
