# Peer review of "Ultrathin Flexible Encapsulation Materials Based on Al2O3/Alucone Nanolaminates for Improved Electrical Stability of Silicon Nanomembrane-Based MOS Capacitors"

_micromachines, 2023, doi:10.3390/mi15010041_

Round 1
Reviewer 1 Report
Comments and Suggestions for Authors
Dear Authors,
Thank you for submitting this interesting study on ALD layers for enhancing the performance of flexible electronics. Please find below suggested revisions to address before publication.
Section 2
Line 84 Define “BOX layer”
Figure 1 This Figure should label the layer thicknesses and drawn to scale if possible to give the reader a better idea of the structure.
Figure 1 Explain in the text why the bending curve radii (85, 110.5, 77.5. 38.5) and strains were chosen.
If possible, provide a cross-sectional SEM image of the layer and device structure.
Explain why you are using 3.5 dyads of ALD. Why not some other number of layers?
Section 3
Line 120 Incomplete sentence “Hence, the pristine……”
Figure 2 Explain in the text the significance of the purple and yellow colors of the coated and non-coated devices. How do you know the edges and sides are covered?
Section 4
The heading for Section 4 is too long. Please make a more concise section heading.
Line 130 Define “PI” adhesive tapes. Polyimide? Please spell it out.
Do you have any photos of the devices in the bending apparatus and bending molds? That would be useful to include to show the reader how you performed the experiment. Please include a photo.
Explain how you measured capacitance. Such as what equipment, point probe, data acquisition, etc…
Every experimental measurement has uncertainty. Please state and/or show the measurement uncertainty in Fig. 3.
Fig 4 requires error/uncertainty bars.
Fig 5 a and b also requires error bars.
Fig 6 a requires a statement of uncertainty/error and b also requires error bars.
Fig 7 a and b requires error bars.
Fig 8 Error/uncertainty
Conclusions
Explain why and how much better the multi-layered structure is than a single thicker layer of ALD.
Provide a statement on what further research is required to advance the application of these coatings. In other words, what should we do next?
Provide a quantitative value on the improvement/retainment of performance with the ALD coatings.
Author Response
Dear Reviewer 1: (Please see the figures in PDF attached below)
Comment 1. (1) Line 84: Define “BOX layer”. (2) Figure 1: This Figure should label the layer thicknesses and drawn to scale, if possible to give the reader a better idea of the structure. (3) Figure 1 Explain in the text why the bending curve radii (85, 110.5, 77.5. 38.5) and strains were chosen. (4) If possible, provide a cross-sectional SEM image of the layer and device structure. (5) Explain why you are using 3.5 dyads of ALD. Why not some other number of layers?
Response 1: We are grateful for the suggestions. As suggested by the reviewer, we label the layer thickness by adding the cross-sectional schematic illustrations in Fig.1 for device structure and 3.5 dyads of ALD films. (Lines 120)
In section 2, we define “BOX layer” as “…burried oxide layer”, in the last part of section 2, we explain the choice of bending curve radii is “…to benchmark against the previous research on Si NM flexible devices”. We are glad that the reviewer is interested in the multilayered structures of ALD. We hope the results of film thickness measured by spectroscopic ellipsometer changed with ALD cycles, can verify the 3.5 dyads of ALD.
For the reason why we chose 3.5 dyads, because in this work, 3.5 dyads of ALD/MLD Al2O3/alucone nanolaminates is using 100 cycles of Al2O3 and 10 cycles of alucone, namely, the ratio of ALD/(ALD+MLD) cycles is ~0.9, which is approaching the sensitivity limit of the MOCON Aquatran 1 instrument of WVTR ~1 × 10-4 g / (m2 day), as summarized in Fig.3 of this paper (DOI:10.1063/1.4766731). Thus, it is valuable to investigate this structure on flexible Si NM based devices.
Comment 2. (1) Line 120 Incomplete sentence “Hence, the pristine……” (2) Figure 2 Explain in the text the significance of the purple and yellow colors of the coated and non-coated devices. How do you know the edges and sides are covered?
Response 2: We apologize for the careless problems in the original manuscript. We correct it immediately in the text as (Lines 135) “…Hence, the electrical characteristics of unencapsulated and Al2O3/alucone encapsulated MOSCAPs in the pristine state were used as baselines for further comparisons of strain-related characteristics.”
We are grateful that the reviewer noticed the significance of the purple and yellow colors of the coated and non-coated devices. We add the demonstration in the text (Lines 123 to 128) “…It is notable from the optical images (Fig. 3(a)) that the anode area of MOSCAPs is yellow before the encapsulation process. It then significantly changes into purple after depositing 3.5 dyads Al2O3/alucone films, and the deposition area can be identified through the arrays of etching holes on the functional layer of Si NM, in which the color of the holes has changed. These changes indicated that the structures of MOSCAPs were seamlessly protected after conformal encapsulation.”
In addition, the color changes provide evidence for the stable interface trap density of the encapsulated device, as demonstrated in the text (Lines 264 to 266), “…On the other hand, the ultrathin Al2O3/alucone balanced the residual strain caused by etching holes on Si NM, which improved the interfacial characteristics compared to unencapsulated MOSCAPs.”
Comment 3. (1) The heading for Section 4 is too long. Please make a more concise section heading. (2) Line 130 Define “PI” adhesive tapes. Polyimide? Please spell it out. (3) Do you have any photos of the devices in the bending apparatus and bending molds? That would be useful to include to show the reader how you performed the experiment. Please include a photo. (4) Explain how you measured capacitance. Such as what equipment, point probe, data acquisition, etc… (5) Every experimental measurement has uncertainty. Please state and/or show the measurement uncertainty in Fig. 3. Fig 4 requires error/uncertainty bars. Fig 5 a and b also requires error bars. Fig 6 a requires a statement of uncertainty/error and b also requires error bars. Fig 7a and b requires error bars. Fig 8 Error/uncertainty
Response 3: We are grateful for the suggestion. We revise the heading for section 4 as (Lines 141) “…Electrical Characteristics at Different Bending Radii”. We define “PI” in section 4.1 as (Lines 145) “…polyimide (PI) adhesive tapes”. We add photos of the device under test as Fig.2 (Lines 120)
We add more details about the experimental setup in Section 2, revised as (Lines 116 to 118) “…C-V and I-V measurements were carried out by Keysight B1500A semiconductor parameter analyzer, which connected to Cascade Microtech EPS 150 manual probe station. All the measurements were performed at room temperature and in a dark box.” Before the discussion, we claim in the text that the measured data is repeatable, and the trends are consistent. That is why we chose representative and significant results for discussion in this work, as (Lines 137 to 140) “…A total of 20 devices were prepared from the same fabrication methods of samples, the test results were repeatable, and the trends on curves were consistent. Although there are slight differences in values, the trends are consistent. Therefore, we selected significant and representative results for discussion.” We believe the measured results were repeatable, the measured C-V curve was repeated three times to verify the repeatability, as shown in the figure below.
Comment 4. (1) Explain why and how much better the multi-layered structure is than a single thicker layer of ALD. (2) Provide a statement on what further research is required to advance the application of these coatings. In other words, what should we do next? Provide a quantitative value on the improvement/retainment of performance with the ALD coatings.
Response 4: Thanks for the reviewer’s precious comments and advice. We add the reason why a multi-layered structure is better than a single layer in the last part of Section 4.3, as follows: (Lines 287 to 293) “…From the device aspect, the ultrathin multilayered structures provide a tortuous path compared to a single layer, preventing the external molecules or gases from penetrating the dielectric layer. This ensures gate oxide reliability when the flexible MOSCAPs endure external bending strains. Compared to the unencapsulated MOSCAPs, the stability of gate leakage current, effective oxide charges, interfacial characteristics, and frequency dispersion are sufficient to evidence that proposal.”
In the future, the investigations can focus on the mechanical stability of ultrathin flexible encapsulation, as follows: (Lines 319 to 322) “…The further development of ultrathin inorganic-organic encapsulation for high-performance, wearable, and implantable electronics needs efforts to excavate scientific issues such as bringing the stretchable designs as a form factor with ultrathin inorganic-organic encapsulation.” As for the quantitative value on the improvement of performance with the ALD coatings, the authors are interested in the investigations on the ALD Al2O3 films for improvement of the cycling performance and thermal ability of LiNi0.5Co0.2Mn0.3O2 composite electrodes with the capacity retention is 76.8 % after 30 cycles. In addition, the authors highlight the significance of the ALD films as barrier coatings for stretchable electrodes, which is very promising to be employed in the next generation of wearable, implantable, and foldable electronics.
Finally, I would like to thank you sincerely again for your insightful suggestions!

Reviewer 2 Report
Comments and Suggestions for Authors
- The introduction claims that "Physical parameters are comprehensively analyzed in this work including accumulation capacitance (Cacc), minimum capacitance (Cmin), gate leakage current density (Jg), hysteresis voltage (Vhy), effective oxide charge (Neff), oxide trapped charge (Not), interface trap density 24 (Dit), and frequency dispersion (Freq. (ω))". However, these are electrical parameters, not physical parameters of the materials under study.
- In the Introduction it is claimed that there is a daunting challenge limiting degradation of Vt and mobility, but then the focus of this work is CV and current density versus. Critical parameters including accumulation capacitance (Cacc), minimum capacitance (Cmin), gate leakage current density (Jg), and hysteresis voltage (Vhy)”. I don’t see why these parameters are critical; the minimum capacitance is set by the substrate doping level, the accumulationcapacitance by the electrical thickness of the dielectric stack, and the authors do not introduce the rationale for this claim in the introduction.
- The depletion/inversion of the CV is not shown, but commented on as significant.
- The authors claim that the Vfb shift of 60 mV is negligible. However, for the oxide capacitance they quote, this amounts to a charge density of order 1e11 cm-2, which is certainly not negligible.
- Line 120: sentence “Hence, the pristine” not finished.
- Line 147: limited evidence of Dit change is presented; can conductance curves be shown to complement? Also, to justify the features seen in the Dit-Eg results shown (see points below)
- IV curves are shown across 8 decades, whereas the data presented lies within 3; can the range be limited to focus on the data presented, such that the insets are not required.
- The reproducibility of these results is not discussed or presented. How reproducible are the phenomena reported, in particular for the small changes. This weakens the confidence in plotting data as relative change in 4b, 5b, 6b,7b, 8b and 9b
- It is claimed that the hysteresis arises from mobile charge. While that is one possibility, another common effect of hysteresis are pre-existing defects within the dielectric changing their charge state during the measurement itself. Has this been explored, for instance by varying maximum bias and seeing the bias dependence of the delta Vfb?
- The benefits and learning from the changes in minimum capacitance are not clear to this reviewer.
- How is the theoretical Vfb calculated? What values are used for metal work function, substrate doping?
- It is claimed that the Neff value is independent of the voltage range. While that is true in equation 2 from where it is calculated, there is no evidence to suggest that the Vfb value itself is independent of applied bias, as could be expected for a defect level distributed energetically within the dielectric bandgap (as routinely seen for HfO2 stacks).
- How is the surface potential calculated as applied in equation 3?
- The Dit value is plotted over an energy window of ~0.4 eV, and yet shows two peak features, with peak FWHM of <~ 50meV. This suggests a very narrow window energetically for an interface state. Is this consistent with the thermal expansion of a single energy defect, if assumed to follow Fermi-Dirac statistics?
- The frequency dependent difference in maximum capacitance is claimed to be related to “the interfacial property for unencapsulated MOSCAPs is relatively poor under strains possibly due to the permeated gas or molecules in the air degrading the interface of the high-k dielectric/Si channel structures”. This reviewer has never seen a change in accumulation capacitance attribute to interface states
Overall, I feel that this article presents a series of observations, but in my opinion lacks a discussion section to tie all the results together. Plotting the results as relative change (eg in Dit), without any comment or consideration of variability renders many of the conclusions as open to debate.
Comments on the Quality of English Language
The level if english is inadequate in this article; the article is not straightforward to read (to a native speaker), and there are numerous errors
Author Response
Dear Reviewer 2: (Please see the figures in PDF attached below)
Comment 1. The introduction must be improved and explain why these electrical parameters are critical. “Electrical parameters” not “physical”.
Response 1: We agree with the comment and add the sentences in the introduction to explain the reason why we investigate these electrical parameters, as the following: (Lines 59 to 69) “…On the one hand, to achieve flexible, high-performance, high-speed, low-power dissipation metal-oxide-semiconductor transistors, high-quality gate oxide is required to realize strong gate control capabilities. Low sub-threshold swing (SS), high current switching ratio (Ion/Ioff), high transconductance (gm), and controllable threshold voltage are critically needed. To obtain higher yield and robust device reliability, investigations should focus on high-quality gate oxide and packaging layers, which can increase yield and improve device reliability. Variating these parameters with strain is essential for enhancing electrical stability and reliability in wearable, foldable, stretchable, and implantable devices. On the other hand, the investigations on encapsulated thin film transistors lack the mechanisms-related evidence to verify the effectiveness of improvement on electrical characteristics at bending strains.” The wrong demonstration was revised as (Lines 21) “electrical parameters”
Comment 2. The depletion/inversion of the CV is not shown, but commented on as significant.
Response 2: We are grateful for the suggestion. We revised the Fig.3(a) and Fig.4 to show depletion and inversion. (Lines 121 and Lines 149)
Comment 3. The authors claim that the Vfb shift of 60 mV is negligible. However, for the oxide capacitance they quote, this amounts to a charge density of order 1e11 cm-2, which is certainly not negligible.
Response 3: We have modified this expression throughout the text according to the comment. (Lines 130) “…In addition, the flatband voltage (Vfb) shifted from -0.76 V to -0.70 V after encapsulation.”
Comment 4. Line 120: sentence “Hence, the pristine” not finished.
Response 4: We apologize for the careless problems in the original manuscript. We correct it immediately in the text as (Lines 135) “…Hence, the electrical characteristics of unencapsulated and Al2O3/alucone encapsulated MOSCAPs in the pristine state were used as baselines for further comparisons of strain-related characteristics.”
Comment 5. Line 147: limited evidence of Dit change is presented; can conductance curves be shown to complement? Also, to justify the features seen in the Dit-Eg results shown (see points below)
Response 5: Thanks for the reviewer’s opinion. We use the Terman method to extract Dit values. Terman developed and used the high frequency capacitance method for determining interface trap capacitance. In this work, we use the measured C-V curve at 100kHz. In the high frequency capacitance method, capacitance is measured as a function of gate bias with the frequency fixed at a high enough value so that interface traps don’t respond.
The interface traps follow very slow changes in gate bias as the MOS capacitor is swept from accumulation to inversion. They cause the high frequency C-V curve to stretch out along the gate bias. The ideal C-V curve is calculated for the same doping density and oxide thickness but without interface traps. As the representative curve shown below provides evidence of the Dit extraction method in this work.
Comment 6. IV curves are shown across 8 decades, whereas the data presented lies within 3; can the range be limited to focus on the data presented, such that the insets are not required.
Response 6: We are incredibly grateful to the reviewer for pointing out this problem. We have revised the Fig.6(a) as (Lines 195)
Comment 7. The reproducibility of these results is not discussed or presented. How reproducible are the phenomena reported, in particular for the small changes. This weakens the confidence in plotting data as relative change in 4b, 5b, 6b,7b, 8b and 9b
Response 7: Thanks for the reviewer’s precious comments and advice. We believe the measured results were repeatable, the measured C-V curve was repeated three times to verify the repeatability, as shown in the figure below. Meantime, we explain this comment in the last Section 3 (Lines 137 to 140) “…A total of 20 devices were prepared from the same fabrication methods of samples, the test results were repeatable, and the trends on curves were consistent. Although there are slight differences in values, the trends are consistent. Therefore, we selected significant and representative results for discussion.”
Comment 8. It is claimed that the hysteresis arises from mobile charge. While that is one possibility, another common effect of hysteresis are pre-existing defects within the dielectric changing their charge state during the measurement itself. Has this been explored, for instance by varying maximum bias and seeing the bias dependence of the delta Vfb?
Response 8: We sincerely appreciate the reviewer’s suggestion. According to the reviewer’s comment, we have provided more details to describe the possible reasons. At first, the encapsulation structure is fabricated through atomic layer deposition and molecular layer deposition. The procedure includes high-energy ion bombardment, which may cause the difference in the electrical properties of encapsulated MOS capacitors. Secondly, the might introduced defects is possible to deteriorate the electrical properties at the bending strains. Thirdly, if the pre-existing defects within the dielectric, it is included in the electrical property at an unbent state, which is undoubtedly not associated with the relative changes at bending strains. At last, because the dielectric in this work being very thin (10 nm), we didn’t apply more voltage on the device in case the device failed.
Most importantly, we think this topic is exciting and we are delighted to implement a series of related works on it. In the meantime, we add the sentences in Section 4.2, as follows: (Lines 210 to 214)“On the other hand, the investigation focuses on the relative change. The possible pre-existing defects in the oxide layer are included as the baseline for encapsulated and unencapsulated MOSCAPs at the original state. Thus, the Vhy-related changes are more likely to be defined as process-related defects, gradually influencing the Vhy at compressive and tensile strains.”
Comment 9. The benefits and learning from the changes in minimum capacitance are not clear to this reviewer.
Response 9: We are grateful for the suggestion. As suggested by the reviewer, we add the sentences in Section 4.2 to explain the benefits of investigation on Cmax and Cmin, as follows: (Lines 163 to 166) “…At an unbent state, 3.5 dyads Al2O3/alucone encapsulation doesn’t deteriorate the electrical performance. It is meaningful to investigate the variations on the dielectric layer at bending strains. The maximum capacitance denotes oxide capacitance, and the minimum capacitance is influenced by semiconductor capacitance.” As investigated in Fig.5(a) and (b), the tensile strain has more influence on the Cmin.
Comment 10. How is the theoretical Vfb calculated? What values are used for metal work function, substrate doping?
Response 10: Thank you for the suggestion. The theoretical Vfb is calculated by
the metal work function is 3.8 eV for Ti, the substrate doping is 1.93×1016 cm-3.
Comment 11. It is claimed that the Neff value is independent of the voltage range. While that is true in equation 2 from where it is calculated, there is no evidence to suggest that the Vfb value itself is independent of applied bias, as could be expected for a defect level distributed energetically within the dielectric bandgap (as routinely seen for HfO2 stacks).
Response 11: The authors appreciate that the reviewer pointed out the problem. We delete the wrong statement in the revised text. The comments on this high-k dielectric are very pertinent. In addition, Neff is effective oxide charge number density, Not is oxide trapped number density.
Comment 12. How is the surface potential calculated as applied in equation 3?
Response 12: Thanks for the reviewer’s opinion. The surface potential (Vs) is calculated as the following relation:
Comment 13. The Dit value is plotted over an energy window of ~0.4 eV, and yet shows two peak features, with peak FWHM of <~ 50meV. This suggests a very narrow window energetically for an interface state. Is this consistent with the thermal expansion of a single energy defect, if assumed to follow Fermi-Dirac statistics?
Response 13: The authors sincerely appreciate the reviewer’s suggestion. The Dit extraction procedure is carefully checked. We found that the fitting method causes the problem of peak features. To verify this issue, we use another fitting method of Gaussian functions to derive the deltaVg vs Vs curve. The Dit vs Eg distribution is reasonable. We comprehensively re-calculated all the results and checked the energy locations. The final results are exhibited in Fig.9 (Lines 240)
Based on the new results, the authors re-written the demonstration and discussion part in Section 4.3. (Lines 242 to 266) “…At the unbent state, the distribution of Dit values monotonically decreased with the energy level for encapsulated and unencapsulated MOSCAPs. With changes in bending strains, the slope of the Dit distribution for unencapsulated MOSCAPs is gradually smoother than the unbent state. However, the Dit distribution for encapsulated MOSCAPs is stable at the applied bending strains. The increase in Dit values significantly demonstrates that the interfacial characteristics deteriorated under bending strains. Yet, 3.5 dyads of Al2O3/alucone ultrathin encapsulation films stabilized the interfacial characteristics at bending strains. Specifically, to compare the relative changes in Dit, the energy level of Ec-E = 0.35 eV was selected as the reference because the changes at that energy level tended to be steady on the Dit distribution. Extracted Dit values divided by the Dit values in the unbent state (Dit/Dit,o) at various bending strains are shown in Fig. 9(b). The Dit values for MOSCAPs without encapsulation are more susceptible to being affected by the tensile strains. The results showed that the Dit of the unencapsulated MOSCAP at Ec-E = 0.35 eV is 0.96×1012 eV-1 cm-2 of unbent state and increased 93.1 % to achieve 1.86×1012 eV-1 cm-2 at +0.33 %. Moreover, the variations are as significant as +9.7 % to +13.8 % for the other three bending strains. In contrast, the Dit of the encapsulated MOSCAP at Ec-E = 0.35 eV maintains steady within the variations of -6.4 % to +11.4 %. The strain-related Dit distributions for Al2O3/alucone encapsulated, and unencapsulated MOSCAPs are comparatively analyzed to prove that the ultrathin encapsulation multilayers are promising to stabilize the interfacial characteristics at bending strains. On the one hand, it may be attributed to the Al2O3/alucone nanolaminates having the ability to prevent ions in the air into the dielectric/Si NM interface from deterioration of electrical performances. On the other hand, the ultrathin Al2O3/alucone balanced the residual strain caused by etching holes on Si NM, which improved the interfacial characteristics compared to unencapsulated MOSCAPs.”
Comment 14. The frequency dependent difference in maximum capacitance is claimed to be related to “the interfacial property for unencapsulated MOSCAPs is relatively poor under strains possibly due to the permeated gas or molecules in the air degrading the interface of the high-k dielectric/Si channel structures”. This reviewer has never seen a change in accumulation capacitance attribute to interface states
Response 14: We agree with the comment and delete the sentence in the revised manuscript. To summarize the result of frequency dispersion, we add the demonstration as (Lines 280) “…It verified that the 3.5 dyads Al2O3/alucone nanolaminates efficiently promoted the stability and reliability of the dielectric layer for Si NM based flexible MOSCAPs.”
Comment 15. Overall, I feel that this article presents a series of observations, but in my opinion lacks a discussion section to tie all the results together. Plotting the results as relative change (eg in Dit), without any comment or consideration of variability renders many of the conclusions as open to debate.
Response 15: We agree with the comment and have rewritten the discussion part in Section 4.3, as follows: (Lines 283 to 305) “…Based on the observations of changes in electrical properties for 3.5 dyads Al2O3/alucone ultrathin films encapsulated and unencapsulated MOSCAPs, the results evoke robust evidence on the mechanisms between Si NM based MOSCAPs and ultrathin inorganic-organic encapsulating materials. The encapsulated Si NM-based MOSCAPs could withstand more compressive and tensile strains than unencapsulated MOSCAPs. From the device aspect, the ultrathin multilayered structures provide a tortuous path compared to a single layer, preventing the external molecules or gases from penetrating the dielectric layer. This ensures gate oxide reliability when the flexible MOSCAPs endure external bending strains. Compared to the unencapsulated MOSCAPs, the stability of gate leakage current, effective oxide charges, interfacial characteristics, and frequency dispersion are sufficient to evidence that proposal. From the fabrication aspect, the ALD/MLD procedure truly has more advantages in precise thickness control. However, the ion bombardment with high energy causes process-related issues in this work, and the variations in hysteresis voltage change with dynamic bending strains. It requires more considerations for designing anode stacks with efficient thickness and the scalable layout of encapsulated devices. From the fracture strain aspect, the mechanical bending strain leads to the microcracks in the multilayered device structures. The physical microcracks influence the electrical property with complex mechanisms. Here, the electro-mechanical analysis on 3.5 dyads Al2O3/alucone nanolaminates encapsulated MOSCAPs highlighted the strain-related electrical parameters such as oxide charges and interface trapped charges, potential tools providing theoretical evidence and practical results to electronic design automation (EDA) designers. It is helpful to develop strain-insensitive, flexible transistors with reliable encapsulation structures.”
Comment 16. The level if english is inadequate in this article; the article is not straightforward to read (to a native speaker), and there are numerous errors.
Response 16: We apologize for the language problems in the original manuscript. The language presentation was improved with assistance from a native English speaker with appropriate research background. The language of the revised paper has been polished in the revised manuscript.
Finally, I would like to thank you sincerely again for your insightful suggestions!

Round 2
Reviewer 2 Report
Comments and Suggestions for Authors I've quickly reviewed the paper and see that most of the changes I recommended have been implemented, as thoroughly as can be expected in the turnaround time from the authors, and overall the quality has been improved.